# Distinct Clinicopathological Features and Prognostic Values of High-, Low-, or Non-Expressing HER2 Status in Colorectal Cancer

**DOI:** 10.3390/cancers15020554

**Published:** 2023-01-16

**Authors:** Zehua Wu, Yi Cheng, Huaiming Wang, Dian Liu, Xiaoxing Qi, Chao Wang, Yuanzhe Zhang, Yuting Zhang, Runkai Cai, Hong Huo, Jianwei Zhang, Yue Cai, Weiwei Li, Huabin Hu, Yanhong Deng

**Affiliations:** 1Department of Medical Oncology, The Sixth Affiliated Hospital, Sun Yat-Sen University, Guangzhou 510655, China; 2Guangdong Provincial Key Laboratory of Colorectal and Pelvic Floor Disease, The Sixth Affiliated Hospital, Sun Yat-Sen University, Guangzhou 510655, China; 3Department of Gastrointestinal Surgery, The First Affiliated Hospital of Shantou University Medical College, Shantou 515000, China; 4Department of Lymphoma and Abdominal Radiotherapy, Hunan Cancer Hospital and the Affiliated Cancer Hospital of Xiangya School of Medicine, Central South University, Changsha 410013, China; 5Department of Medical Oncology, The Third Affiliated Hospital of Xinxiang Medical University, Xinxiang 453002, China; 6Department of Pathology, The Sixth Affiliated Hospital, Sun Yat-Sen University, Guangzhou 510655, China

**Keywords:** ERBB2, immunohistochemistry, colorectal neoplasms, clinicopathological feature

## Abstract

**Simple Summary:**

In recent years, the antibody-drug conjugate (ADC) of Human Epidermal Growth Factor Receptor 2 (HER2) has been found to play an important role in some HER2-negative cancers as in it did in HER2-positive patients. Therefore, a more detailed and suitable classification of HER2 is needed. Our study revealed that HER2-low colorectal cancer tumors did not show an intermediate state of HER2 expression in clinicopathology and prognosis. HER2-low colorectal cancer tumors are like HER2-zero tumors, with a lower proportion of perineural invasion, lower tumor stage and more RAS/BRAF mutation, compared with HER2-high tumors. Multivariate analysis and propensity score matching also revealed that HER2-high expression was an independent prognostic factor of disease-free survival. Our study indicated that the routine examination of HER2 status is needed in early-stage colorectal cancer.

**Abstract:**

The encouraging effects of HER2-ADC in patients with HER2-low expression cancers indicated the classical classifications based on positive and negative HER2 might no longer be suitable. However, the biology and prognosis of colorectal cancer patients with different HER2 expression status were still not clear. This is a multi-center retrospective study that included patients with histologically confirmed colorectal cancer and determined HER2 status who received radical surgical resection. HER2 immunohistochemistry (IHC) 1+ and IHC 2+ groups were combined and defined as a HER2-low group because of the concordance of clinicopathological characteristics. As compared with the HER2-high group, both the HER2-zero and the HER2-low group had less tumor with perineural invasion (14.3%, 13.1% vs. 31.6%, *p* = 0.001 and *p* < 0.001), less stage III disease (41.8%, 39.9% vs. 56.1%, *p* = 0.044 and *p* = 0.022), more RAS/BRAF mutation (52.1%, 49.9% vs. 19.5%, *p* < 0.001 and *p* < 0.001) and better disease-free survival (DFS) (3y-DFS rate of 78.7%, 82.4% vs. 59.3%, *p* < 0.001 and *p* < 0.001). Multivariate analysis and propensity score matching also revealed that HER2-high expression was an independent prognostic factor of DFS. In conclusion, our study revealed that HER2-low colorectal cancer tumors are close to HER2-zero tumors, but different from HER2-high tumors. The routine examination of HER2 IHC is needed in early-stage colorectal cancer.

## 1. Introduction

The Erb-B2 receptor tyrosine kinase, also known as HER2, is a proto-oncogene located on chromosome band 17q21, which leads to tumor occurrence and development [1,2]. HER2 is an established therapeutic target in breast cancer, gastric cancer, as well as in colorectal cancer [3,4,5,6]. Since targeted therapy of HER2 is only approved for the treatment of metastatic colorectal cancer, HER2 testing is recommended for patients with metastatic colorectal cancer but not early-stage colorectal cancer [7].

Furthermore, the prognostic value of HER2 expression in colorectal cancer remained unclear and has been controversial in early-stage colorectal cancer. Several studies suggest that there is no association between HER2 protein expression and prognosis in colorectal cancer [8,9,10,11,12]. However, in the studies mentioned above, the criteria of defining HER2 status was adopted, which, used in breast cancer and HER2 IHC staining 2+ together with staining 3+, were classified as positive [8,9,10,11,12]. These criteria and classification were discordant with National Comprehensive Cancer Network (NCCN) guidelines [7]. Whereas some studies have shown no association, other studies have found that HER2 expression might be associated with prognosis in colorectal cancer. Dong II Park et al. found that HER2 overexpression was associated with poor 3-year (70.8% vs. 83.7%) and 5-year survival rates (55.1% vs. 78.3%); it was also found to be independently related to survival by multivariate analysis [13]. However, the conclusion should be cautious because the limited size of 137 patients of the whole study and the criteria of defining HER2 status, which was also adopted from breast cancer [13]. Another study from Huang et al. showed that patients with HER2 positivity had worse survival rates in stage III colorectal cancer [14]. The antibody used and the ambiguous HER2 diagnostic criteria were not those recommended in NCCN, which was the major limitation of this study [14]. All in all, the conclusion about the prognostic value of HER2 expression in colorectal cancer is still controversial.

In addition, the clinical efficacy of the blockade of HER2, including monoclonal antibodies such as trastuzumab and pertuzumab, and tyrosine kinase inhibitors, such as lapatinib, neratinib and tucatinib, were only observed to be effective in the HER2 overexpression/amplification (IHC score 3+ or IHC 2+/in situ hybridization [ISH] positive) tumors [3,4,5,6]. Therefore, clinicians mainly focus on the detection of HER2 overexpression/amplification. The HER2-targeted ADC, such as Trastuzumab deruxtecan (T-DXd; formerly DS-8201a), also shows encouraging effects on some patients with HER2-negative tumors [15,16]. The classical classifications based on positive and negative HER2 will no longer be suitable for future research and treatment. Currently, HER2 expression in breast cancer has been updated as HER2-high (defined as IHC score 3+ or 2/ISH positive), HER2-low (defined as IHC score 1 or 2/ISH negative) and HER2-zero (IHC score 0) [17]. This prompted us to explore the pathologic features and clinical outcome of HER2-zero, HER2-low and HER2-high subsets in colorectal cancer. As prognosis in patients with metastatic disease can be influenced by a variety of factors, we analyzed the characteristic and prognostic value of HER2 expression in 2783 patients with histologically confirmed stage I-III colorectal cancer in multiple centers.

## 2. Materials and Methods

### 2.1. Study Population

This was a multi-center retrospective study that included patients with histologically confirmed colorectal cancer and determined HER2 status who received radical surgical resection between April 2013 and April 2022 at The Sixth Affiliated Hospital of Sun Yat-Sen University, The First Affiliated Hospital of Shantou University Medican College, Hunan Cancer Hospital, The Third Affiliated Hospital of Xinxiang Medical University. Patients with middle or lower third rectal cancer (with a distance of less than 10 cm from the anal verge), stage IV disease, tumors other than carcinoma, and incomplete curative resection (R1 or R2 resection) were excluded. The clinicopathological characteristics, gene status as well as survival data were collected from hospital records. Ethical approval was given by the Institutional Review Boards of The Sixth Affiliated Hospital of Sun Yat-Sen University.

### 2.2. Analysis of HER2 Protein Expression by IHC

Formalin-Fixed and Parrffin-Embedded (FFPE) tumors were stained for the HER2 protein. Following the manufacturers’ instructions, HER2 expression by IHC was performed manually using the HercepTest antibody (Dako A/S Glostrup, Denmark) and automatically on the automated BenchMark Ultrasystem using the VENTANA 4B5 antibody. The HER2 protein expression level is scored as 0, 1+, 2+ and 3+ by IHC; a score of 3+ was considered a strong HER2 expression, 2+ was considered equivocal, 1+ was considered faint, and 0 was considered as not expressed. The details of the IHC test are according to the HERACLES diagnostic criteria [18]. Because the anti-HER2 targeted therapy was only applied in patients with metastatic colorectal cancer, further fluorescence in situ hybridization (FISH) was not routinely carried out in patients with early-stage disease.

### 2.3. Mismatch Repair (MMR) Status Determination

The MMR status was tested through the analysis of MMR protein expression by IHC. Deficient MMR (dMMR) phenotype tumors were defined as exhibiting the loss of expression of 1 or more MMR proteins by IHC, including MLH1, MSH2, MSH6, PMS2, and proficient MMR (pMMR) was defined as the intact expression of all MMR proteins [19].

### 2.4. Gene Mutation Detection

Genomic DNA extraction from FFPE tumor resection samples was performed using an EZgene Tissue gDNA miniprep kit (Cat no: GD2211, Biomiga, Hangzhou, China). KRAS (exons 2, 3, 4), NRAS (exons 2, 3, 4), BRAF (exon 15, V600E mutations), and PIK3CA (exon 9 and 20) mutations were evaluated by bidirectional sequencing using an ABI Prism 3 500 DX Genetic Analyzer (Applied Biosystems, Foster City, CA, USA).

### 2.5. Treatment and Follow-Up

All patients in this study were treated by radical surgery, and most of the stage III and high-risk stage II patients underwent postoperative adjuvant chemotherapy for 6 months in the perioperative course of patients undergoing surgery. The follow-up strategy included physical examination, serum carcinoembryonic antigen (CEA), and thoraco-abdominopelvic computed tomography scans every 3 to 6 months in the first 3 years and every 6 months in the following 2 years. The follow-up data were updated in August 2022.

### 2.6. Propensity Score Matching

We performed propensity score matching to reduce the baseline characteristics imbalance between patients with HER2-high and those with HER2-zero and HER2-low. A multivariable logistic regression model, including factors considered to be related with prognosis, was constructed to generate propensity scores. Factors included in the model included: age at diagnosis (<60 or ≥60 years), rectal cancer or not, initial bowel obstruction, differentiation (poorly or well/moderately), pathologic T stage (T4 or T1–3), vascular invasion and/or lymphatic infiltration, perineural invasion, MMR status (dMMR or pMMR), lymph node metastasis, tumor deposits, and number of lymph nodes excised (<12 or ≥12). Patients with HER2-high were matched to those with HER2-zero or HER2-low in a 1:4 ratio, respectively, and patients with HER2-low were matched to those with HER2-zero in a 1:1 ratio according to a greedy nearest-neighbor matching algorithm. Baseline characteristics were compared using standardized differences between the propensity score-matched groups. A standard difference (SMD) of less than 0.1 was considered as a balance between groups [20].

### 2.7. Statistical Analysis

Categorical variables were compared by dint of the Chi-square test or Fisher’s exact test. DFS was defined as survival without local or metastatic recurrence, second primary cancer, or death from any cause. Variables with *p* values less 0.05 in univariate analysis were included in the multivariate analyses. The propensity score matching was performed using the package MatchIt, and together with other statistical information, was inputted in R, version 4.0.3 (R Foundation, Vienna, Austria). The statistical significance was set at a *p*-value < 0.05.

## 3. Results

### 3.1. Patient Characteristics

A total of 2768 patients with HER2 IHC status was identified (Figure 1). The median age at diagnosis was 60 (range, 17 to 92), with 59.4% of the patients being male. Using the eighth edition of the Tumor Node Metastasis (TNM) classification of malignant tumors, pathological TNM staging was determined, including 231 (8.3%) stage I, 1391 (50.3%) stage II, and 1146 (41.4%) stage III individuals. Additionally, 1729 patients had complete data for KRAS, NRAS, and BRAF status.

### 3.2. Distribution of HER2 Expression of Score 0, 1, 2, 3

The whole cohort was analyzed according to the HER2 IHC score. The number of specimens with different immunostaining scores are listed below: 1680 (score 0), 648 (score 1), 383 (score 2) and 57 (score 3). Representative images of HER2 score of 0,1,2,3 were present in Figure 2a–d. The clinicopathological patient characteristics are shown in Appendix A for the four groups of patients. Overall, the status of differentiation grade, initial bowel obstruction, vascular invasion and/or lymphatic infiltration, perineural invasion, pathological T stage, and RAS/BRAF gene mutation were significantly different among the four groups. However, the status of age, sex, tumor location, lymph node metastasis and MMR were all similar (Appendix A).

### 3.3. Distinct Clinical Features of HER2-Zero, Low, High Colorectal Cancer

To assess similarities and differences of clinicopathological characteristics across all HER2 expression groups, we firstly compared the HER2 IHC 1+ group and the IHC 2+ group. There was no statistical difference in any baseline characteristics (all *p*-values for comparisons > 0.3) (Appendix A). We next defined a HER2 IHC score of 0 as HER2-zero, IHC 1+ and IHC 2+ as HER2-low, and IHC 3+ as HER2-high.

As compared with the HER2-high group, both the HER2-zero and the HER2-low group had less tumor with perineural invasion (14.3% vs. 31.6%, *p* = 0.001 and 13.1% vs. 31.6%, *p* < 0.001) and less stage III disease (41.8% vs. 56.1%, *p* = 0.044 and 39.9% vs. 56.1%, *p* = 0.022) (Table 1 and Figure 3a,b). We next assessed the similarity and difference of the HER2-zero and HER2-low group. As expected, the proportion of perineural invasion and stage III disease were both similar between the groups (14.3% vs. 13.1%, *p* = 0.415 and 41.8% vs. 39.9%, *p* = 0.328 and Figure 3a,b). However, other characteristics do not exactly coincide between the two groups. Differences in initial obstruction, differentiation, T stage and vascular invasion and/or lymphatic infiltration could also be observed (Table 1).

Furthermore, less RAS/BRAF mutation was found in the HER2-high group than those in the HER2-zero (19.5% vs. 52.1%, *p* < 0.001) and HER2-low group (19.5% vs. 49.9%, *p* < 0.001), while no significant difference was found between the HER2-zero (52.1% vs. 49.9%, *p* = 0.415) and the HER2-low group (Figure 3c). Together, these findings suggest differences in clinical features between HER2-zero, HER2-low, and HER2-high colorectal cancers.

### 3.4. Prognostic Relevance of HER2 Groups in the Overall Cohort and Subgroups

At the time of data cutoff, the median follow-up was 22.4 months for the overall cohort. The 3y-DFS rate of the HER2-high group, HER2-low group and HER2-zero were 59.3% (95% CI, 45.7–72.9), 82.4% (95% CI, 80.0–84.8), and 78.7% (95% CI, 76.7–80.7), respectively. The DFS was significantly shorter in the HER2-high group, as compared with the HER2-zero (*p* < 0.001) and HER2-low group (*p* < 0.001). As expected, there was no significant difference when comparing the DFS between the HER2-zero and the HER2-low group (*p* = 0.226) (Figure 4a).

A similar difference was also observed in the patients of RAS/BRAF wild-type. Patients with HER2-low (3y-DFS:82.1% [95% CI, 77.5–86.7] vs. 48.7% [95% CI, 30.1–67.3], *p* < 0.001), HER2-zero (3y-DFS:81.7% [95% CI, 78.3–85.1] vs. 48.7% [95% CI, 30.1–67.3], *p* < 0.001) both had significantly longer DFS, as compared with the HER2-high expression group (Figure 4b). However, no obvious difference was observed in HER2-zero, HER2-low and HER2-high groups with regard to the RAS/BRAF mutated subgroup (Figure 4c).

A multivariate analysis revealed that HER2-high expression was an independent prognostic factor of DFS (HR, 2.05; 95% CI, 1.33–3.16; *p* = 0.001) (Table 2). In addition to HER2 expression, age, rectal cancer, initial obstruction, grade of differentiation, T stage, vascular invasion and/or lymphatic infiltration, perineural invasion, lymph node metastasis, and MMR status were significant prognostic factors in the multivariate analysis (Table 2).

### 3.5. Propensity Score Matching

At 1:4 propensity score matching, 57 patients with HER2-high were matched to the patients with HER2-zero and HER2-low, respectively. As shown in Appendix A, after propensity score matching, standardized differences for most of the included covariates were less than 0.1. After matching, a significant reduced DFS was observed in the HER2-high group as compared with the HER2-zero (3y-DFS:59.3% [95% CI, 45.7–72.9] vs. 71.6% [95% CI, 65.5–77.7], *p* = 0.040) and HER2-low expression group (3y-DFS:59.3% [95% CI, 45.7–72.9] vs. 79.3% [95% CI, 73.8–84.8], *p* = 0.002) (Appendix A). A multivariate analysis also revealed that HER2-high expression was an independent factor of DFS after matching with the HER2-zero group (HR, 1.72; 95% CI, 1.04–2.84; *p* = 0.035) and the HER2-low group (HR, 2.30; 95% CI, 1.36–3.91; *p* = 0.002). Next, another propensity score matching was also performed, where 1032 patients with HER2-low expression were matched to the patients with HER2-zero at a 1:1 ratio (Appendix A and Appendix A). A multivariate analysis revealed that HER2-low expression was no independent prognostic factor after matching with HER2-zero (HR, 0.84; 95% CI, 0.67–1.06; *p* = 0.14) (Table 3).

## 4. Discussion

The purpose of this study was to explore the clinical characteristics and prognostic values of colorectal cancer with different HER2 status. We included 2768 individuals diagnosed with colorectal cancer in multiple centers. The major finding of the present study was that the HER2-low group resembled to the HER-zero expression group in terms of biological behavior and prognosis. In contrast, HER2-high expression had distinct different clinicopathological features and prognosis as compared with the HER2-zero and HER2-low expression groups. HER2 IHC tests should be considered routinely in clinical practice with regard to early-stage colorectal cancer to guide prognosis and treatment. Compared with multiple studies on the prognostic value of HER2 expression in colorectal cancer, our study adopted the diagnostic criteria of HER2 expression as recommended by NCCN guidelines and enrolled many colorectal cancer patients from multiple centers, which did not show larger interpatient variability. Previous research has found that the prevalence of HER2 overexpression is higher in RAS/BRAF wild-type colorectal tumors [6,21,22]. Thus, according to NCCN guidelines, HER2 testing is not indicated if the tumor is already known to have a RAS/BRAF mutation. Patients with RAS/BRAF mutations were not candidates for dual-targeted therapy [7]. In our study, we also explored the prognostic value of HER2 expression under different RAS/BRAF gene status. To our knowledge, this was the first study of clinicopathologic features, gene alterations, and the prognostic value of HER2-zero and HER2-low in early-stage colorectal cancer.

Compared with the other two groups, HER2-high tumors are more associated with perineural invasion and a higher TNM stage. HER2-high tumors showed less RAS/BRAF mutation frequency, but there was no significant difference between HER2-low and HER2-zero tumors. Although a few differences of clinical and pathological characteristics remained between HER2-low and HER2-zero groups, no difference of prognosis between HER2-low and HER2-zero was observed. HER2-low expression tumors are close to HER2-zero tumors, but different from HER2-high tumors. Our findings are consistent with similar studies in metastatic colorectal cancer, which demonstrated that HER2-low metastatic colorectal cancer is similar to HER2-zero colorectal cancer in terms of prognostic value and molecular landscape [23]. These results appear to differ from studies in breast cancer, which showed that tumors with HER2-low expression have different biological and clinical characteristics and prognosis than HER2-zero tumors [24]. HER2-low breast tumors were characterized by a higher frequency of hormone positive and PIK3CA mutations, a lower frequency of TP53 mutations, and the pCR rate of neoadjuvant therapy when compared to HER2-zero tumors [25]. In our study, the prognosis value of HER2-high in the RAS/BRAF wild-type cohort was consistent with the overall cohort. However, in the RAS/BRAF mutation subgroup, there is no significant difference in DFS between HER2-high, HER2-low, and HER2-zero tumors. The RAS/BRAF mutation is very common in colorectal cancer and plays an important role in tumorigenesis and progression [26]. Approximately 5% of metastatic colorectal cancer is driven by the amplification of HER2 [27]. The results of our study suggest that the driving effect of HER2 is inconsistent under different RAS/BRAF status, which is the major pathway downstream of HER2 [26]. It is well known that HER2 overexpression can present an intratumor and intertumoral heterogeneity pattern [28]. The treatment effect of trastuzumab, a monoclonal antibody that binds and inhibits HER, was different in the HER2 overexpression/amplification in breast cancer, gastric cancer, and colorectal cancer [4,29,30]. Although the DESTINY-Break04 study showed that T-DXd significantly prolonged the progression-free survival and overall survival in patients with advanced breast cancer with low HER2 expression [31], the response rate to DS-8201 is limited in colorectal cancer patients with HER2 low expression, as reported by the DESTINY-CRC01 study [32]. The activation of RAS/RAF signaling may be contributing to the low efficacy of T-DXd in HER2-low metastatic colorectal cancers [27] because HER2-low metastatic colorectal cancers are reported with more RAS relative pathway enrichment [23]. Another possible reason could be intratumor heterogeneity. A study showed that the proportion of HER2-expressed cells in HER2-low colorectal cancer was only 20%, with a maximum of 60% [23].

Our study had inherent limitations. Firstly, incomplete documentation or missing data are a limitation of retrospective research. Secondly, the FISH test was not conducted in this study, so our cohort did not distinguish the HER2 IHC 2+ population, but we do not believe that this limits our findings. A few HER2 2+/FISH+ patients were included in the HER2-low group in our study, and they were assumed to have a poorer prognosis if they resembled HER2-high. If participants with HER2 2+/FISH+ were excluded from the HER2-low group, the prognosis difference between HER2-low and HER2-high groups may be more obvious, which indicates that our data was weakly affected by the FISH result. The data showed that most of the colorectal cancers with HER2 IHC 2+ are negative by FISH. The incidence of HER2 amplification by FISH accounts for only 5% to 11% of cases with a score of 2+ by IHC [18]. Therefore, our analysis was not affected by the FISH positive rate in HER2 2+. In addition, only RAS/BRAF gene status was analyzed. Due to the lack of other gene mutation landscapes and other omic data, which restricted the further exploration of different HER2 subgroups.

## 5. Conclusions

In conclusion, our study revealed that HER2-low colorectal cancer tumors are close to HER2-zero tumors, but different from HER2-high tumors. Tumors with HER2-zero and HER2-low expression were associated with better DFS in non-metastatic colorectal cancer compared with those with HER2-high expression. The routine examination of HER2 IHC is needed in early-stage colorectal cancer and will help to guide precise treatment in clinical practice.

## Figures and Tables

**Figure 1 cancers-15-00554-f001:**
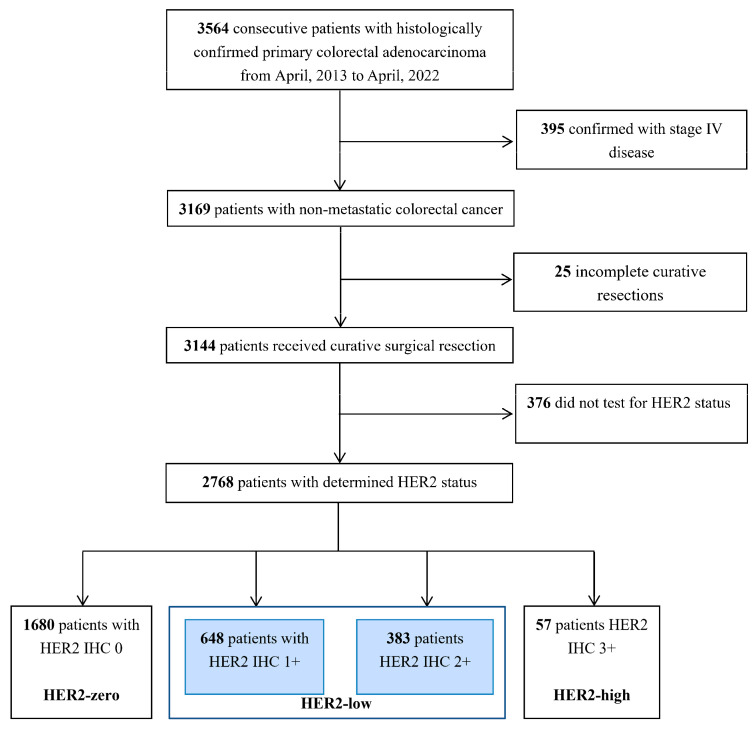
Diagram of the study design.

**Figure 2 cancers-15-00554-f002:**
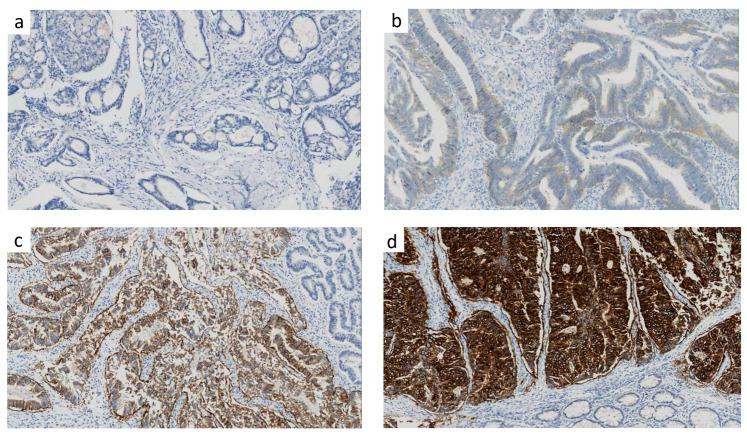
Representative images of HER2 proteins. Representative images of HER2 IHC score of 0 (**a**), 1 (**b**), 2 (**c**), 3 (**d**); 10×; IHC, immunohistochemistry.

**Figure 3 cancers-15-00554-f003:**
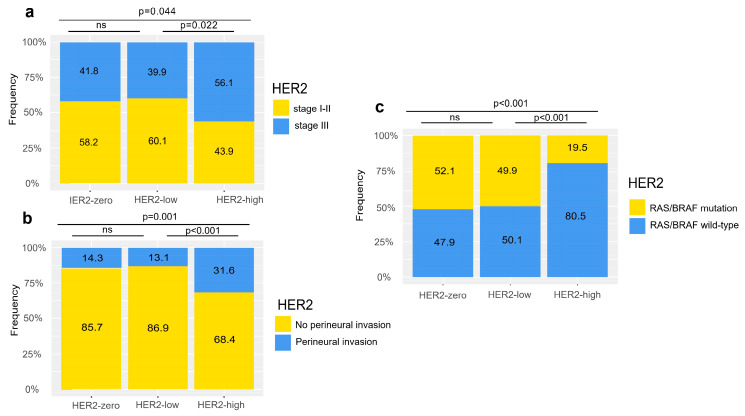
HER2 subgroups had different clinical features, including (**a**) pathological stage, (**b**) perineural invasion, and (**c**) RAS/BRAF mutation.

**Figure 4 cancers-15-00554-f004:**
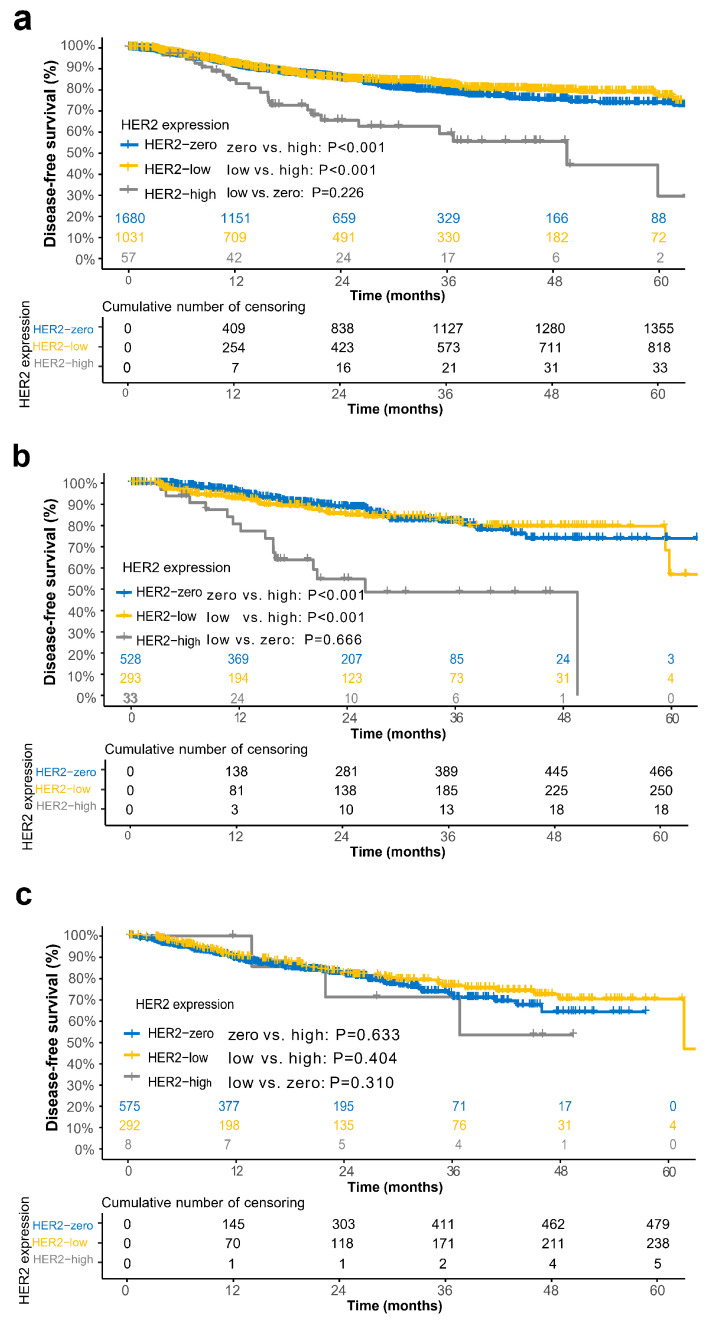
Kaplan-Meier survival analysis for disease-free survival. Comparison of HER2-zero, HER2-low and HER2-high groups for the overall cohort, (**a**) for RAS/BRAF wild-type cohort, (**b**) and RAS/BRAF mutation cohort (**c**). *p* values are from the Cox proportional hazards regression model.

**Table 1 cancers-15-00554-t001:** Baseline clinicopathological characteristics of the overall cohort.

Characteristics	All the Population, *n* = 2768	HER2-Zero Group, *n* = 1680	HER2-Low Group, *n* = 1031	HER2 High Group, *n* = 57	*p*
	No. (%)	No. (%)	No. (%)	No. (%)	
Age, years					
<60	1325 (47.9%)	804 (47.9%)	490 (47.5%)	31 (54.4%)	ns
≥60	1443 (52.1%)	876 (52.1%)	541 (52.5%)	26 (45.6%)	
Gender					
Female	1125 (40.6%)	695 (41.4%)	406 (39.4%)	24 (42.1%)	ns
Male	1643 (59.4%)	985 (58.6%)	625 (60.6%)	33 (57.9%)	
Grade of differentiation					zero vs. low, *p* < 0.001
Well- or moderately	2373 (85.7%)	1395 (83.0%)	925 (89.7%)	53 (93.0%)	zero vs. high, ns
Poorly	395 (14.3%)	285 (17.0%)	106 (10.3%)	4 (7.0%)	low vs. high, ns
Primary tumor site					
Left (splenic flexure, descending colon, sigmoid colon, and rectum)	1685 (60.9%)	1010 (60.1%)	638 (61.9%)	37 (64.9%)	ns
Right (cecum, ascending colon, hepatic flexure, and transverse colon)	1083 (39.1%)	670 (39.9%)	393 (38.1%)	20 (35.1%)	
Rectal cancer					
No	2719 (98.2%)	1652 (98.3%)	1012 (98.2%)	55 (96.5%)	ns
Yes	49 (1.8%)	28 (1.7%)	19 (1.8%)	2 (3.5%)	
Initial bowel obstruction					zero vs. low, *p* < 0.001
No	2631 (95.1%)	1578 (93.9%)	1000 (97.0%)	53 (93.0%)	zero vs. high, ns
Yes	137 (4.9%)	102 (6.1%)	31 (3.0%)	4 (7.0%)	low vs. high, ns
Vascular invasion and/or lymphatic infiltration					zero vs. low, *p* = 0.016
No	2471 (89.3%)	1483 (88.3%)	941 (91.3%)	47 (82.5%)	zero vs. high, ns
Yes	297 (10.7%)	197 (11.7%)	90 (8.7%)	10 (17.5%)	low vs. high, *p* = 0.045
Perineural invasion					zero vs. low, ns
No	2375 (85.8%)	1440 (85.7%)	896 (86.9%)	39 (68.4%)	zero vs. high, *p* = 0.001
Yes	393 (14.2%)	240 (14.3%)	135 (13.1%)	18 (31.6%)	low vs. high, *p* < 0.001
No. of lymph nodes excised					
<12	284 (10.3%)	160 (9.5%)	118 (11.4%)	6 (10.5%)	ns
≥12	2484 (89.7%)	1520 (90.5%)	913 (88.6%)	51 (89.5%)	
Pathologic T stage					zero vs. low, *p* < 0.001
T1–T3	2349 (84.9%)	1467 (87.3%)	835 (81.0%)	47 (82.5%)	zero vs. high, ns
T4	419 (15.1%)	213 (12.7%)	196 (19.0%)	10 (17.5%)	low vs. high, ns
Lymph node metastasis					zero vs. low, ns
No	1770 (63.9%)	1064 (63.3%)	678 (65.8%)	28 (49.1%)	zero vs. high, *p* = 0.041
Yes	998 (36.1%)	616 (36.7%)	353 (34.2%)	29 (50.9%)	low vs. high, *p* = 0.016
Tumor deposit					zero vs. low, ns
No	2283 (82.5%)	1394 (83.0%)	848 (82.3%)	41 (71.9%)	zero vs. high, *p* = 0.047
Yes	485 (17.5%)	286 (17.0%)	183 (17.7%)	16 (28.1%)	low vs. high, ns
Pathologic N stage					zero vs. low, ns
N0	1622 (58.6%)	977 (58.2%)	620 (60.1%)	25 (43.9%)	zero vs. high, *p* = 0.044
N1–2	1146 (41.4%)	703 (41.8%)	411 (39.9%)	32 (56.1%)	low vs. high, *p* = 0.022
Mismatch repair status					
Proficient	2429 (87.8%)	1472 (87.6%)	902 (87.5%)	55 (96.5%)	ns
Deficient	339 (12.2%)	208 (12.4%)	129 (12.5%)	2 (3.5%)	
RAS/BRAF mutation					zero vs. low, ns
No	854 (49.4%)	528 (47.9%)	293 (50.1%)	33 (80.5%)	zero vs. high, *p* < 0.001
Yes	875 (50.6%)	575 (52.1%)	292 (49.9%)	8 (19.5%)	low vs. high, *p* < 0.001
Missing values					
Neoadjuvant therapy					
No	2765 (99.9%)	1677 (99.8%)	1031 (100%)	57 (100%)	ns
Yes	3 (0.1%)	3 (0.2%)	0 (0%)	0 (0%)	
Adjuvant therapy					
No	1377 (49.7%)	828 (49.3%)	526 (51.0%)	23 (40.4%)	ns
Yes	1391 (50.3%)	852 (50.7%)	505 (49.0%)	34 (59.6%)	

**Table 2 cancers-15-00554-t002:** Univariate and multivariate Cox proportional hazards regression model for disease-free survival of the overall cohort.

Variable	No. Patients	No. Events	Univariate Analysis		Multivariate Analysis	
			HR (95% CI)	*p*	HR (95% CI)	*p*
Total	2768	405				
Age, years						
<60	1325	188	1	0.02	1	0.033
≥60	1443	217	1.26 (1.04–1.53)		1.24 (1.02–1.52)	
Gender						
Female	1125	166	1	0.734		
Male	1643	239	1.03 (0.85–1.26)			
Grade of differentiation						
Well- or moderately	2373	333	1	0.017	1	0.041
Poorly	395	72	1.36 (1.06–1.76)		1.33 (1.01–1.73)	
Pathologic T stage						
T1–T3	2349	289	1	<0.001	1	<0.001
T4	419	116	2.27 (1.83–2.82)		1.9 (1.52–2.37)	
Vascular invasion and/or lymphatic infiltration						
No	2471	331	1	<0.001	1	0.022
Yes	297	74	2.04 (1.59–2.63)		1.38 (1.05–1.81)	
Perineural invasion						
No	2375	286	1	<0.001	1	
Yes	393	119	2.8 (2.26–3.46)		1.98 (1.58–2.5)	<0.001
No. of lymph nodes excised						
≥12	284	57	1	0.007	1	0.121
<12	2484	348	1.47 (1.11–1.95)		1.26 (0.94–1.67)	
Initial bowel obstruction						
No	2631	368	1		1	<0.001
Yes	137	37	2.77 (1.97–3.89)	<0.001	2.04 (1.44–2.89)	
Lymph node metastasis						
No	1770	183	1	<0.001	1	0.005
Yes	998	222	1.87 (1.54–2.28)		1.36 (1.1–1.68)	
Tumor deposit						
No	2283	283	1	<0.001	1	0.058
Yes	485	122	1.83 (1.48–2.27)		1.25 (0.99–1.57)	
Primary tumor site						
Left (splenic flexure, descending colon, sigmoid colon, and rectum)	1685	267	1	0.063		
Right (cecum, ascending colon, hepatic flexure, and transverse colon)	1083	138	0.82 (0.67–1.01)			
Rectal cancer						
No	2719	390	1	0.016	1	0.031
Yes	49	15	1.88 (1.12–3.15)		1.78 (1.06–3.01)	
Mismatch repair status						
Proficient	2429	384	1	<0.001	1	0.005
Deficient	339	21	0.39 (0.25–0.6)		0.52 (0.33–0.83)	
Adjuvant therapy						
No	1377	141	1	<0.001		
Yes	1391	264	1.54 (1.25–1.89)			
HER2 status						
Zero and low	2711	383	1	<0.001	1	0.001
High	57	22	2.58 (1.68–3.96)		2.05 (1.33–3.16)	

Abbreviation: CI, confidence interval; HR, hazard ratio.

**Table 3 cancers-15-00554-t003:** Multivariate Cox proportional hazards regression model for disease-free survival in propensity score matching cohorts.

Variable	HER2-Zero vs. HER2-High		HER2-Low vs. HER2-High		HER2-Zero vs. HER2-Low	
	HR (95% CI)	*p*	HR (95% CI)	*p*	HR (95% CI)	*p*
Total						
Age, years						
<60	1	0.045	1	0.191	1	0.126
≥60	1.62 (1.01–2.61)		1.41 (0.84–2.37)		1.20 (0.95–1.51)	
Rectal cancer						
No	1	0.047	1	0.526	1	0.055
Yes	2.62 (1.01–6.76)		1.40 (0.49–4.00)		1.86 (0.99–3.52)	
Initial bowel obstruction						
No	1	0.434	1	0.007	1	<0.001
Yes	1.40 (0.60–3.23)		2.71 (1.31–5.58)		2.46 (1.53–3.96)	
Grade of differentiation						
Well- or moderately	1	0.684	1	0.933	1	0.509
Poorly	0.79 (0.26–2.40)		1.04 (0.39–2.78)		1.14 (0.78–1.66)	
Pathologic T stage						
T1–T3	1	0.010	1	0.122	1	<0.001
T4	2.11 (1.19–3.73)		1.78 (0.86–3.69)		1.81 (1.40–2.33)	
Vascular invasion and/or lymphatic infiltration						
No	1	0.008	1	0.989	1	0.290
Yes	2.27 (1.24–4.15)		1.01 (0.46–2.18)		1.21 (0.85–1.72)	
Perineural invasion						
No	1	0.002	1	0.991	1	<0.001
Yes	2.24 (1.34–3.74)		1.00 (0.57–1.74)		2.26 (1.73–2.95)	
Lymph node metastasis						
No	1	0.631	1	0.228	1	0.024
Yes	1.14 (0.67–1.95)		1.43 (0.80–2.54)		1.32 (1.04–1.69)	
Tumor deposit						
No	1	0.682	1	0.188	1	0.056
Yes	0.89 (0.52–1.54)		1.47 (0.83–2.61)		1.3 (0.99–1.70)	
No. of lymph nodes excised						
≥12	1	0.736	1	0.988	1	0.141
<12	1.14 (0.53–2.44)		0.99 (0.47–2.09)		1.27 (0.92–1.75)	
Mismatch repair status						
Proficient	1	0.277	1	0.996	1	0.037
Deficient	0.32 (0.04–2.48)		0 (0-Inf)		0.58 (0.35–0.97)	
HER2 status						
Zero	1	0.035			1	0.140
Low			1	0.002	0.84 (0.67–1.06)	
High	1.72 (1.04–2.84)		2.30 (1.36–3.91)			

## Data Availability

The data presented in this study are available from the corresponding author upon reasonable request.

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
