# Peer review of "Distinct Clinicopathological Features and Prognostic Values of High-, Low-, or Non-Expressing HER2 Status in Colorectal Cancer"

_cancers, 2023, doi:10.3390/cancers15020554_

Round 1
Reviewer 1 Report
This is an interesting manuscript where the authors aim to assess the HER2 expression status in colorectal cancer and its association between clinicopathology and prognosis in CRC patients. The manuscript is quite comprehensive, and the results are also interesting.
Nonetheless, I have several remarks:
1. I recommend the authors rewrite the simple summary section as it doesn’t provide the overall concept/summary of the research question that the authors want to address in their paper.
2. Reframe the sentence “Score of 0, 1, 2, 3 were identified in 1680 (60.7%), 648 (23.4%), 383 (13.8%) and 57 (2.1%)” on line number 159-160 (page number 4).
3. I recommend the authors provide high-resolution images for figure number 3 and 4.
4. I recommend that Table 1 should fit into a single page so that the reader doesn’t have to go back and forth to look into the details, hence, I would recommend preparing the table accordingly.
5. Reframe the sentence “A significantly shorter DFS was observed in the HER2-high group, as compared with HER2-zero and HER2-low group, with a 3y-DFS rate of 59.3% (95%CI, 45.7- 72.9), 78.7% (95%CI, 76.7-80.7), 82.4% (95%CI, 80.0-84.8) (p<0.001 and p<0.001)” on line number 198-200 (page number 9).
6. I understand that the authors mentioned that, “HER2 high expression is an independent factor of poor prognosis in non-metastatic colorectal cancer independent of the RAS/RAF mutation status”. Are there any studies/reports that show any link between RAS/RAF mutation status and HER2 expression?
7. The authors must avoid repetition of information; this will make the article more comprehensive, and the language of the paper should be checked by a native English speaker to make the paper fluent so that the readers don’t lose interest while going through it.
Reviewer 2 Report
In the research article “Distinct clinicopathological features and prognostic values of high-, low-, or non-expressing HER2 status in colorectal cancer” Zehua Wu et al analyzed IHC HER2 status from the perspective of the prognostic factor. The authors focus to compare HER2-zero and the HER2-low group versus HER2-high. The author’s suggestions that routine examination of HER2 IHC is needed in early-stage colorectal cancer are very reasonable. The authors did not mention it, but HER2 testing is now included in the National Comprehensive Cancer Network (NCCN) colon cancer treatment guidelines, which highlights the importance of that type of data collection. The present study is a multi-center retrospective study.
Major concerns:
1 Introduction and Discussion sections require major revision. Multiple studies in the field with the same or close aim have been published. Authors need to use them to justify the need for the present study, to compare outcomes of their own results, and to discuss them in depth.
Authors may want to discuss some of those articles:
1. Kavanagh DO, Chambers G, O'Grady L, Barry KM, Waldron RP, Bennani F, Eustace PW, Tobbia I. Is overexpression of HER-2 a predictor of prognosis in colorectal cancer? BMC Cancer. 2009;9:1.
2. Marx AH, Burandt EC, Choschzick M, Simon R, Yekebas E, Kaifi JT, Mirlacher M, Atanackovic D, Bokemeyer C, Fiedler W, Terracciano L, Sauter G, Izbicki JR. Heterogenous high-level HER-2 amplification in a small subset of colorectal cancers. Hum Pathol. 2010;41:1577–1585.
3. Wang XY, Zheng ZX, Sun Y, Bai YH, Shi YF, Zhou LX, Yao YF, Wu AW, Cao DF. Significance of HER2 protein expression and HER2 gene amplification in colorectal adenocarcinomas. World J Gastrointest Oncol. 2019 Apr 15;11(4):335-347. doi: 10.4251/wjgo.v11.i4.335. PMID: 31040898; PMCID: PMC6475672.
4. Li Q, Wang D, Li J, Chen P. Clinicopathological and prognostic significance of HER-2/neu and VEGF expression in colon carcinomas. BMC Cancer. 2011;11:277.
5. Nathanson DR, Culliford AT 4th, Shia J, Chen B, D'Alessio M, Zeng ZS, Nash ZM, Gerald W, Barany F, Paty PB. HER 2/neu expression and gene amplification in colon cancer. Int J Cancer. 2003;105:796–802.
6. Park DI, Kang MS, Oh SJ, Kim HJ, Cho YK, Sohn CI, Jeon WK, Kim BI, Han WK, Kim H, Ryu SH, Sepulveda AR. HER-2/neu overexpression is an independent prognostic factor in colorectal cancer. Int J Colorectal Dis. 2007;22:491–497.
7. Ooi A, Takehana T, Li X, Suzuki S, Kunitomo K, Iino H, Fujii H, Takeda Y, Dobashi Y. Protein overexpression and gene amplification of HER-2 and EGFR in colorectal cancers: an immunohistochemical and fluorescent in situ hybridization study. Mod Pathol. 2004;17:895–904.
8. Al-Kuraya K, Novotny H, Bavi P, Siraj AK, Uddin S, Ezzat A, Sanea NA, Al-Dayel F, Al-Mana H, Sheikh SS, Mirlacher M, Tapia C, Simon R, Sauter G, Terracciano L, Tornillo L. HER2, TOP2A, CCND1, EGFR and C-MYC oncogene amplification in colorectal cancer. J Clin Pathol. 2007;60:768–772.
2 Authors focused on Antibody Drag Conjugate therapy in the introductions, discussion, and even in the Abstract, but ADC is not the subject of this study. It is not clear why the authors discussed HER2-ADC therapy in breast and gastric cancer in the Introduction. It is a very weak justification of the present study and confuses the reader. A possible association between HER2 expression and ADC efficiency may be speculated in the discussion but should have stronger support from previously published studies.
3 The major finding of the study is that the HER2-low group resembled to the HER-zero expression group in terms of biological behavior and prognosis. However, the authors found differences in those two groups in initial obstruction, differentiation, T stage, and vascular invasion and/or lymphatic infiltration. Moreover, some of these characteristics at the same time had no significant differences with HER2-high. The authors did not explain and/or discuss these results in depth.
4 Statements below need to be clarified:
Collectively, the landscape of cancer genes and mutational processes are different in different types of tumors. The status of HER2 as a driver oncogene is different in different types of tumors. The treatment effect of trastuzumab, a monoclonal antibody that binds and inhibits HER, was different in HER2 overexpression/amplification breast cancer, gastric cancer, and colorectal cancer [20–22].
Moreover, in the Discussion there is an overstatement without reference support: “However, different from results of gastric cancer and breast cancer, clinical trials of HER2-ADCs in CRC had not found any benefit in HER2-negative CRC.”
Minor:
1 Figure 3, and Figure 4 – Nonreadable, please increase Font size
2 In methods section 2.4 used two times
3 Correct double spaces, double dots, and spaces before dots.
4 Figure 2 description is redundant, please simplify.
Round 2
Reviewer 2 Report
Please, proofread the text and check the typos, and misspellings (e.g. lines 254, 263, etc).